# A Survey on Knowledge Graphs for Healthcare: Resources, Applications, and Promises

**Hejie Cui** [1]  **Jiaying Lu** [1]  **Shiyu Wang** [1*]  **Ran Xu** [1*]  **Wenjing Ma** [1*]  **Shaojun Yu** [1*]  **Yue Yu** [2*]  **Xuan Kan** [1*]
**Chen Ling** [1]  **Joyce Ho** [1]  **Fei Wang** [3]  **Carl Yang** [1]

## Abstract

Healthcare knowledge graphs (HKGs) have emerged as a promising tool for organizing medical knowledge in a structured and interpretable way, which provides a comprehensive view of medical concepts and their relationships. However, challenges such as data heterogeneity and limited coverage remain, emphasizing the need for further research in the field of HKGs. This survey paper serves as the first comprehensive overview of HKGs. We summarize the pipeline and key techniques for HKG construction (i.e., from scratch and through integration), as well as the common utilization approaches (i.e., model-free and model-based). To provide researchers with valuable resources, we organize existing HKGs[1] based on the data types they capture and application domains, supplemented with pertinent statistical information. In the application section, we delve into the transformative impact of HKGs across various healthcare domains, spanning from fine-grained basic science research to high-level clinical decision support. Lastly, we shed light on the opportunities for creating comprehensive and accurate HKGs in the era of large language models, presenting the potential to revolutionize healthcare delivery and enhance the interpretability and reliability of clinical prediction.

## 1. Introduction

A knowledge graph (KG) is a data structure that captures the relationships between different entities and their at-

tributes (Ji et al., 2021; Nicholson & Greene, 2020). KG models and integrates data from various sources, including structured and unstructured data, and has been studied to support a wide range of applications such as search engines (Wang et al., 2019a), recommendation systems (Wang et al., 2019b; Zhou et al., 2020), and question answering (Lin et al., 2019; Yasunaga et al., 2021; Yan et al., 2021; Kan et al., 2021). Particularly for healthcare, KG facilitates an interpretable representation of medical concepts such as drugs and disease, which enables context-aware insights and enhances clinical research, decision-making, and healthcare delivery (Santos et al., 2022; Chandak et al., 2023).

On the data side, Healthcare knowledge graphs (HKGs) are usually built on the landscape from complex medical systems such as electronic health records, medical literature, clinical guidelines, and patient-generated data (Bouayad et al., 2017; Rajkomar et al., 2018). However, these data resources are often heterogeneous and distributed, making it challenging to integrate and analyze them effectively (Mehta & Pandit, 2018). The data heterogeneity can also lead to incomplete or inconsistent data representations within HKGs, limiting their usefulness for downstream healthcare tasks (Dash et al., 2019). Additionally, the current use of domain-specific knowledge graphs may result in limited coverage and granularity of the knowledge captured across different levels, hindering the ability to identify correlations and relationships between medical concepts from multiple domains. These challenges underscore the need for continued research on HKGs to fully realize their potential.

On the model side, HKG can be constructed from scratch or through the integration of existing dataset resources, where many key steps such as entity and relation extraction can be optimized depending on natural language processing tools and algorithms. Recent progress in general domain knowledge extraction has leveraged advances in pre-trained large language models, (*i.e.* BERT (Devlin et al., 2019), GPT-3 (Brown et al., 2020)). These models revolutionize the field of natural language processing, enabling the efficient and effective integration of heterogeneous medical data from various sources. The use of pre-trained models has also allowed for the development of more accurate and comprehensive

---

[*]Equal contribution [1]Department of Computer Science, Emory University, Atlanta, GA [2]School of Computational Science and Engineering, Georgia Institute of Technology, Atlanta, GA [3]Department of Population Health Sciences, Weill Cornell Medicine, New York, NY, USA. Correspondence to: Carl Yang <j.carlyang@emory.edu>.

---

[1]The resource is available at https://github.com/lujiaying/Awesome-HealthCare-KnowledgeBase

medical ontologies and taxonomies (Zhang et al., 2021a; Yu et al., 2020; Wang et al., 2021a; Zeng et al., 2021), allowing for the construction of unprecedentedly comprehensive and fine-grained KGs for healthcare (Xu et al., 2020).

A comprehensive and fine-grained healthcare knowledge graph holds the potential to revolutionize healthcare across various levels (Gyrard et al., 2018; Santos et al., 2022; Li et al., 2020). At the micro-scientific level, HKGs can help researchers identify new phenotypic and genotypic correlations and understand the underlying mechanisms of disease (Hassani-Pak & Rawlings, 2017), leading to more targeted and effective treatments (Seneviratne et al., 2021; Chandak et al., 2023). At the clinical care level, HKGs can be used to develop clinical decision support systems that provide clinicians with relevant information, improving clinical workflows and patient outcomes (Eberhardt et al., 2012; Castaneda et al., 2015). Therefore, conducting an extensive survey of the existing literature on healthcare knowledge graphs becomes an indispensable roadmap and invaluable resource for constructing a comprehensive HKG that can drive transformative advancements in healthcare.

To the best of our knowledge, this survey paper represents the first comprehensive overview of healthcare knowledge graphs (HKGs). The content overview of the survey is depicted in Figure 1, providing a visual summary of the key aspects discussed. We delve into the construction pipelines of HKGs, including both building from scratch and integration approaches, and highlight the key techniques employed in HKG construction. Additionally, we explore two common utilization methods of HKGs, namely model-free and model-based approaches (Section 2). In Section 3, we compile a comprehensive summary of existing HKG resources across various applications, serving as a valuable reference for researchers interested in utilizing or building upon HKGs. Furthermore, we meticulously investigate the literature on mainstream health applications, offering an in-depth overview of the diverse use cases of HKGs in healthcare (Section 4). Finally, we address the unique challenges associated with HKGs and discuss promising research directions, particularly in leveraging large language models to enhance their potential (Section 5). This survey paper targets a wide range of audience, including researchers, practitioners, clinicians, and other experts in healthcare, medical informatics, data science, and artificial intelligence.

## 2. Backgrounds

### 2.1. HKG Definition

A healthcare knowledge graph (HKG) is a domain-specific knowledge graph designed to capture medical concepts such as drugs, diseases, genes, phenotypes, and so on, and their relationships in a structured and semantic way.

### 2.2. HKG Construction

Healthcare knowledge graphs can be constructed from scratch or through the integration of existing data resources.

**Constructing HKGs from Scratch.** A multi-step pipeline, as in Figure 2, is used to construct HKGs from scratch.

1. The first step is to identify the scope and objectives. In most cases, researchers develop a schema (Guarino et al., 2009; Blagec et al., 2022) or use existing schemas (Guarino et al., 2009; Ashburner et al., 2000; Schriml et al., 2012; Bard et al., 2005) to serve as the formal and explicit specification of a domain, thus ensuring consistent, coherent, and aligned domain knowledge. Unlike the general domain KG, utilizing schemas is a common practice in HKG construction.
2. Secondly, researchers gather data from various sources, including medical literature, clinical trials, and patient-generated data. It's essential to ensure the quality and consistency of the data and to remove any identifiable information to protect patient privacy.
3. The third step is to extract and transform the data into a structured format. This step involves identifying medical entities and creating relationships between them via specialized biomedical Natural Language Processing (NLP) tools (Song et al., 2021; Xing et al., 2020; Hahn & Oleynik, 2020).
4. Next, researchers map the entities and relationships to the chosen ontologies with the help of thesauruses (Bodenreider, 2004) or terminologies (Donnelly et al., 2006; Hirsch et al., 2016). This step ensures that the knowledge graph is interoperable with other healthcare systems and facilitates data integration.
5. Until now, an initial KG has been built. The next step is to populate the KG to infer missing links between entities. This inference can be done using graph databases (Wang et al., 2020) or link prediction models (Bordes et al., 2013; Lu & Yang, 2022).
6. The final step is to continuously update and validate the KG to ensure accuracy and relevance. This step involves incorporating new data and knowledge, refining the schema, and evaluating the quality of the KG.

**Constructing HKGs by Integration.** Considering significant efforts have been paid to construct and curate HKGs from scratch, it is promising to integrate these data resources to avoid repetitive work. Healthcare KG integration (also called Healthcare KG fusion) refers to the processing of merging two or more HKGs into a single, more comprehensive graph (Himmelstein et al., 2017; Su et al., 2023; Youn et al., 2022). The integration process is challenging because different HKGs may use different terminologies, schemas, or data formats. To address these challenges, researchers have developed various techniques and algorithms for knowledge graph fusion, including ontology

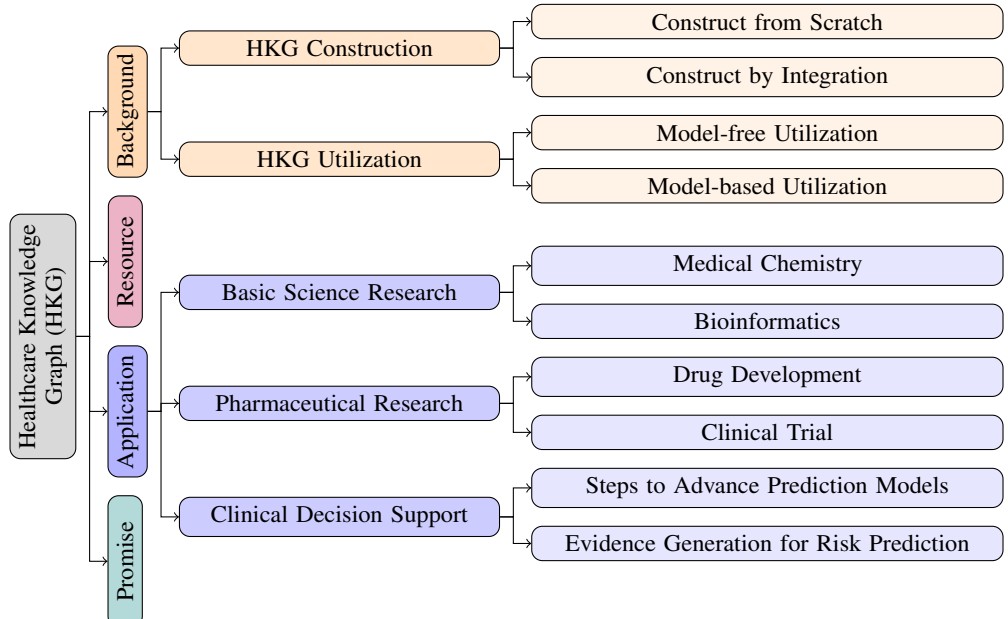

*Figure 1.* Overview of healthcare knowledge graph in this survey.

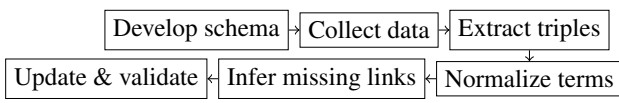

*Figure 2.* The pipeline of constructing HKGs from scratch.

matching (Faria et al., 2014; He et al., 2022), schema alignment (Suchanek et al., 2011; Maaroufi et al., 2014), entity resolution (Bachman et al., 2018; Hu et al., 2021), and conflicts resolution (Ma et al., 2023). These methods aim to identify and reconcile the differences between KGs.

**Techniques for HKG Constructions.** Traditionally, each step of HKG construction involves one specially designated model. For instance, Hidden Markov Models and Recurrent Neural Networks are widely used for healthcare named entity recognition, relation extraction, and other sequence tagging tasks, while Translational Models and Graph Neural Networks are used for HKG completion and conflicts resolution tasks. Recently, large language models (LLMs) have shown great utility to serve as a uniform tool for constructing KGs (Ye et al., 2022). Several key steps of constructing KGs, such as named entity recognition (Liang et al., 2020; Chen et al., 2023a; Huang et al., 2022; Liu et al., 2022a), relation extraction (Zhuang et al., 2022; Lu et al., 2022; Yang et al., 2022), entity linking (De Cao et al.; Mrini et al., 2022; Cho et al., 2022), and KG completion (Geng et al., 2022; Saxena et al., 2022; Xie et al., 2022; Shen et al., 2022), have been successfully tackled by these large foundation models. Early explorations of construction HKG with large foundation models show that healthcare entity normalization (Zhang et al., 2023; Agrawal et al., 2022),

healthcare entity recognition (Fries et al., 2021; Hu et al., 2023), healthcare entity linking (Zhu et al., 2022a), and healthcare knowledge fusion (Lu et al., 2023) can also be performed, without extensive training on expensive healthcare annotated corpus. On the other hand, researchers start to construct KGs under the open-world assumption (Shi & Weninger, 2018; Das et al., 2020; Niu et al., 2021; Li et al., 2022; Lu & Yang, 2022), thus getting rid of the dependency on pre-defined schemas and exhaustive entity&relation normalization. Although open-world KGs greatly increase the coverage, ensuring the quality of extracted knowledge is still an open research challenge, especially for explainable and trustworthy HKGs.

## 2.3. HKG Utilization

**Model-free Utilization.** Various query languages can be used for KGs, such as SPARQL, Cypher, and GraphQL (Wang et al., 2020). These query languages allow users to query healthcare KGs using a standardized syntax, thus enabling users to retrieve, manipulate, and analyze data in a structured and consistent way. More complex applications can be further supported by graph queries. For instance, automatic healthcare question answering can be tackled by Natural Language Question-to-Query (NLQ2Query) approach (Kim et al., 2022), where natural language questions are first translated into executable graph queries and then answered by the query responses. HKGs can also be utilized as an up-to-date and trustworthy augmentation to large language models (LLMs) for many applications. Some pioneering studies (Liu et al., 2022b; Guu et al., 2020; Xu et al., 2023b; Shi et al., 2023) show that retrieved knowl-

edge triples can improve the reliability of LLMs in various knowledge-intensive tasks, by addressing the nonsensical or unfaithful generation. Moreover, KGs can be a useful tool for fact-checking (Tchechmedjiev et al., 2019; Vedula & Parthasarathy, 2021; Mayank et al., 2022) as they provide a structured representation of information that can be used to quickly and efficiently verify the accuracy of claims. Researchers have explored the utility of HKGs in identifying ingredient substitutions of food (Shirai et al., 2021), COVID-19 fact-checking (Mengoni & Yang, 2022), etc.

**Model-based Utilization.** Utilizing HKGs in complex reasoning tasks often involves utilizing machine learning models. HKG embeddings (Yu et al., 2021; Su et al., 2022) have shown great potential to tackle these tasks. In particular, HKG embedding models are a class of machine learning models that aim to learn low-dimensional vector representations, or embeddings, of the entities and relations in a knowledge graph. After obtaining HGK embeddings, they can be plugged into any kind of deep neural network and further fine-tuned toward downstream objectives. On the other hand, symbolic logic models represent another prominent approach for KG reasoning due to their interpretability. More specifically, symbolic reasoning models first mine logical rules from existing knowledge by inductive logic programming (Muggleton, 1992), association rule mining (Galárraga et al., 2013), or Markov logic networks (Kok & Domingos, 2005). These minded rules are used to infer new facts, make logical deductions and answer complex queries. Recently, researchers start to explore combining logical rules into KG embedding to further improve the generalization and performance of HKG reasoning (Alshahrani et al., 2017; Zhu et al., 2022b).

## 3. Resources

In this section, we aim to provide a detailed resource overview of the current state of healthcare knowledge graphs (HKGs), as a reference for researchers and healthcare professionals interested in developing and applying HKGs. A vast range of HKG resources from various domains is organized in Table 1, with attribute information including HKG name, node types, edge types, statistics, and their applications.

## 4. Applications

### 4.1. Basic Science Research

Several previous biological terms can also be considered as knowledge graphs such as ontology (e.g., gene ontology, cell ontology, disease ontology), network (e.g., gene regulatory network), etc. We use the original biological terms as they are more popular according to historical reasons.

### 4.1.1. MEDICINAL CHEMISTRY

Topics related to medicinal chemistry involve drug-drug interactions (DDIs) and drug-target interactions (DTIs), which will be discussed in this section.

**Drug-drug interactions (DDIs)** refer to changes in the actions, or side effects, of drugs when they are taken at the same time or successively (Giacomini et al., 2007). In general, DDIs are a significant contributor to life-threatening adverse events (Su et al., 2022; Pang et al., 2022; Yu et al., 2023), and their identification is one of the key tasks in public health and drug development. The existence of diverse datasets on drug-drug interactions (DDIs) and biomedical KGs has enabled the development of machine learning models that can accurately predict DDIs (Zhong et al., 2023). Yu et al. (2021) develop SumGNN, a model that includes a subgraph extraction module to efficiently extract relevant subgraphs from a KG, a self-attention-based summarization scheme to generate reasoning paths within the subgraph, and a multichannel module for integrating knowledge and data, resulting in significantly improved predictions of multi-typed DDIs. Su et al. (2022) propose DDKG, an attention-based KG representation learning framework that involves an encoder-decoder layer to learn the initial embeddings of drug nodes from their attributes in the KG. Karim et al. (2019) compare various techniques for generating KG embeddings with different settings and conclude that a combined convolutional neural network and LSTM yield the highest accuracy when predicting drug-drug interactions (DDIs). Dai et al. (2021) propose a new KG embedding framework by introducing adversarial autoencoders based on Wasserstein distances and Gumbel-Softmax relaxation for DDI tasks. Lin et al. (2020) develop KGNN that resolves the DDI prediction by capturing drug and its potential neighborhoods by mining their associated relations in KG.

**Drug-target interactions (DTIs)** is just as important as DDIs (Chen et al., 2016). Machine learning models can leverage knowledge graphs constructed from various types of interactions, such as drug-drug, drug-disease, protein-disease, and protein-protein interactions, to aid in the prediction of DTIs. For instance, Li et al. (2023) utilize the KG transfer probability matrix to redefine the drug-drug and target-target similarity matrix, thus constructing the final graph adjacent matrix to learn node representations by VGAE and augmenting them by utilizing dual Wasserstein Generative Adversarial Network with gradient penalty. Zhang et al. (2021c) propose a new hybrid method for DTI prediction by first constructing DTI-related KGs and then employing graph representation learning model to obtain feature vectors of the KG. Wang et al. (2022b) construct a knowledge graph of 29,607 positive drug-target pairs by DistMult embedding strategy, and propose a Conv-Conv module to extract features of drug-target pairs. Ye et al.

*Table 1.* Resource of Existing Healthcare Knowledge Graphs (HKGs).

| Name | Node Types | Edge Types | Statistic | Application |
|---|---|---|---|---|
| HetioNet (Himmelstein & Baranzini, 2015) | 11 (e.g., drug, disease) | 24 (e.g., drug-disease) | #N: 47.0 K, #E: 2.3 M | Medicinal Chemistry |
| DrKG (Ioannidis et al., 2020) | 13 (e.g., disease, gene) | 107 (e.g., disease-gene) | #N: 97 K, #E: 5.8 M | Medicinal Chemistry |
| PrimeKG (Chandak et al., 2023) | 10 (e.g., phenotypes) | 30 (e.g., disease-phenotype) | #N: 129.4 K, #E: 8.1 M | Medicinal Chemistry |
| Gene Ontology[2] (Ashburner et al., 2000) | 3 (e.g., biological process) | 4 (e.g., partOf) | #N: 43 K, #E: 7544.6K | Bioinformatics |
| KEGG[3] (Kanehisa & Goto, 2000) | 16 (e.g., pathway) | 4 (e.g., partOf) | #N: 48.5 M, #E: unknown | Bioinformatics |
| STRING[4] (Szklarczyk et al., 2023) | 1 (e.g., protein) | 4 (e.g., interactions) | #N: 67.6 M, #E: 20 B | Bioinformatics |
| Cell Ontology[5] (Diehl et al., 2016) | 1 (i.e., cell type) | 2 (e.g, subClassOf) | #N: 2.7 K, #E: 15.9 K | Bioinformatics |
| GEFA (Ranjan et al., 2022) | 510 (e.g., kinases) | 2 (e.g., drug-drug) | #N: 0.5 K, #E: 30.1 K | Drug Development |
| Reaction (Li & Chen, 2022) | 2 (e.g., reactant & normal) | 19 (e.g., reaction paths) | #N: 2192.7 K, #E: 932.2 K | Drug Development |
| ASICS (Jeong et al., 2022) | 2 (e.g., reactant & product) | 1 (e.g., reactions) | #N: 1674.9 K, #E: 923.8 K | Drug Development |
| Hetionet (Jeong et al., 2022) | 11 (e.g., biological process) | 24 (e.g., disease–associates–gene) | #N: 47.0 K, #E: 2250.2 K | Drug Development |
| LBD-COVID (Zhang et al., 2021b) | 1 (i.e., concept) | 1 (i.e., SemMedDB relation) | #N: 131.4 K, #E: 1016.1 K | Drug Development |
| GP-KG (Gao et al., 2022) | 7 (e.g., drug) | 9 (e.g., disease–gene) | #N: 61.1 K, #E: 1246.7 K | Drug Development |
| DRKF (Zhang & Che, 2021) | 4 (e.g., drug) | 43 (e.g., drug-disease) | #N: 12.5 K, #E: 165.9 K | Drug Development |
| DDKG (Ghorbanali et al., 2023) | 2 (i.e., drug & disease) | 1 (e.g., drug-disease) | #N: 551, #E: 2.7 K | Drug Development |
| Disease Ontology[6] (Schriml et al., 2012) | 1 (i.e., disease) | 2 (e.g., subClassOf) | #N: 11.2 K, #E: 8.8 K | Clinical Decision Support |
| DrugBank (Wishart et al., 2018) | 4 (e.g., drug, pathway) | 4 (e.g., drug-target) | #N: 7.4 K, #E: 366.0 K | Clinical Decision Support |
| KnowLife (Ernst et al., 2014) | 6 (e.g., genes) | 14 (e.g., gene-diseases) | #N: 2.9 M, #E: 11.4 M | Clinical Decision Support |
| PharmKG (Zheng et al., 2021) | 3 (e.g. diseases) | 3 (e.g. chemical-diseases) | #N: 7601, #E: 500958 | Clinical Decision Support |
| ROBOKOP[7] (Bizon et al., 2019) | 54 (e.g., genes, drugs) | 1064 (e.g. biolink, CHEBI) | #N: 8.6M, #E: 130.4 M | Clinical Decision Support |
| iBKH[8] (Su et al., 2023) | 11 (e.g., anatomy, disease) | 18 (e.g., anatomy-gene) | #N: 2.4 M, #E: 48.2 M | Clinical Decision Support |

(2021b) learn a low-dimensional representation for various entities in the KG, and then integrate the multimodal information via neural factorization machine.

### 4.1.2. BIOINFORMATICS RESEARCH

Knowledge graphs have been in use in Bioinformatics research for quite some time. In a Bioinformatics setting, a knowledge graph is a type of resource that represents biomedical knowledge in a structured and interconnected way. It is a graph-based representation where nodes are biomedical entities (such as mutations, genes, proteins, metabolites, diseases, and biological pathways) and edges are their relationships (such as associations, interactions, regulations)(Nicholson & Greene, 2020).

**Conventional Bioinformatics Resource**: While not explicitly referred to as such, we consider many conventional biomedical resources, such as Gene Ontology (Ashburner et al. (2000)), STRING (Szklarczyk et al. (2023)), KEGG (Kanehisa & Goto (2000)) can be classified as knowledge graphs. These resources have already been extensively utilized and have been shown to make significant advancements in current biomedical research. Gene Ontology (GO) is a knowledge graph that contains information about the functions of genes and their products. It is widely used in gene annotation and functional analysis. STRING is another example of a biomedical knowledge graph. It is a database that contains information about protein-protein interactions (PPIs). STRING integrates information from multiple sources, including experimental data, literature, and databases, to provide a comprehensive view of PPIs. KEGG (Kyoto Encyclopedia of Genes and Genomes) is a knowledge graph that contains information about biological pathways and networks. KEGG integrates information from multiple sources, including genomics, metabolomics,

and systems biology, to provide a comprehensive view of cellular processes.

**Multi-Omics Applications**: In recent years, the field of multi-omics analysis has become increasingly important for understanding biological systems, such as genomics, transcriptomics, proteomics, metabolomics, and epigenomics.

HKGs have been used to identify disease-associated mutations, genes, proteins, and metabolites by integrating multi-omics data with existing biological knowledge. This approach has led to the discovery of novel biomarkers and therapeutic targets for various diseases and interpreting the functional effects of genetic elements. Quan et al, built a comprehensive multi-relational HKG, called AIMedGraph, providing interpretation of impact of genetic variants on disease or treatment (Quan et al.). GenomicsKG (Jha et al., 2019) is an HKG to analyze and visualize multi-omics data. GenomicsKG can be used to improve drug development based on clinical genomics correlations and personalized drug customization in the extended version based on interactive relationships.

**Single-Cell**: Cells are fundamental and essential units of living organisms. With high-throughput sequencing technologies advancing to measure genomic profiles in a single-cell resolution, cell functions (inside cells) and cell-cell interactions (between cells) are revealed (Linnarsson & Teichmann, 2016). The gene regulatory mechanism, visualized by gene regulatory networks (GRNs), plays a crucial role in cell functions, impacting gene expression, cell differentiation, and disease progression. GRNs depict interactions between genes and their regulators and can be largely expanded by mining the whole-genome scale measurements provided by single-cell sequencing data. For example, GRNdb provides detailed regulon and TF-target pairs information from different human and mouse tissues under different conditions

by analyzing existing sequencing data (Fang et al., 2021). GenomicKB integrates existing datasets along with genome annotations and formulates the data into a KG to emphasize the relationships among genomic entries (Feng et al., 2023). On the other hand, cell-cell interactions also help understand cell cycles, cell fate decisions, tissue development, etc. Among several approaches through that cells can interact, cell-cell communication or cell signaling is of the most interest (Eltzschig et al., 2006), cells send signaling molecules called ligands and receive them through receptors located on cell surfaces. Recent advancements in spatial sequencing technologies incorporate colocalization information to better model and score ligand-receptor interactions (LRIs) as shown in recent studies (Liu et al., 2022c; Dimitrov et al., 2022). Although many LRIs databases (Efremova et al., 2020; Shao et al., 2021) have been constructed and applied to infer cell-cell communication, only recently, SpaTalk (Shao et al., 2022) integrates CellTalkDB, KEGG pathways, Reactome and TFs from AnimalTFDB to construct a ligand-receptor-target KG to help improve the inference of cell-cell communication. The success of SpaTalk indicates that correctly utilizing the KG could provide helpful information in cell-cell communication tasks. With the continuous collection of cell type information in Cell Ontology (Diehl et al., 2016), adding cell types as nodes into the biomedical KG could potentially provide biomedical researchers with cell-type-specificity and higher-resolution information as Bioteque does (Fernández-Torras et al., 2022). We hope that a comprehensive KG constructed by mining single-cell sequencing data could help revolutionize the understanding of biological mechanisms in different cells.

## 4.2. Pharmaceutical Research Development

### 4.2.1. DRUG DEVELOPMENT

Drug development identifies novel chemical compounds that can effectively treat or alleviate human diseases. Despite the growing trend of computer-assisted drug discovery (Wang et al., 2022a; Zeng et al., 2022; Pan et al., 2022a; Du et al., 2021; Wang et al., 2022c; Zhang & Zhao, 2021), there remains a key question regarding how to effectively integrate data and extract valuable insights from the vast chemical dataset. To approach this question, KGs have been employed for drug discovery due to various advantages (Zeng et al., 2022): (1) In contrast to traditional methods that capture only one type of relationship, KGs are capable of providing heterogeneous information that includes diverse entities (e.g., scaffolds, proteins and genes); (2) KGs can handle multiple types of relationships between various types of entities, such as drug-target pairs; and (3) KGs can provide unstructured semantic relationships between entities. In such graphs, entities are represented as nodes while their relationships are represented as edges, by which complex relations in biochemical systems can be easily handled.

In general, the field of drug development encompasses two main areas: *drug design* and *drug repurposing*. Drug design creates novel and diverse drug molecules with desirable pharmaceutical properties (Jing et al., 2018; Fu et al., 2021), whereas drug repurposing identifies new uses for existing approved drugs that were originally developed for a different indication (Pan et al., 2022b; Huang et al., 2020).

**Drug Design.** Knowledge Graphs are widely employed in drug design, particularly in the generation of novel molecules that hold promise as potential drug candidates for various diseases (Ranjan et al., 2022; Li & Chen, 2022). Ranjan et al. (2022) utilize Gated Graph Neural Network (GGNN) to generate novel molecules that target the coronavirus (i.e., SARS-CoV-2) (Hasöksüz et al., 2020) and integrate KGs into their approach to reduce the search space. Specifically, KGs were leveraged to discard non-binding molecules before inputting them into the Early Fusion model, thus optimizing the efficiency of the drug design process. In addition to employing deep learning for direct structure design, KGs are also utilized in the analysis of chemical synthesis. Quantitative estimation of molecular synthetic accessibility plays a critical role in prioritizing the molecules generated from generative models. For instance, Li & Chen (2022) utilize reaction KGs to construct classification models for compound synthetic accessibility. By leveraging KGs that capture information about reactions, including reaction types, substrates, and reaction conditions, they are able to train machine learning models that could predict the synthetic accessibility of compounds. Jeong et al. (2022) introduce an intelligent system that integrates generative exploration and exploitation of reaction knowledge base to support synthetic path design.

**Drug Repurposing.** Compared to drug design, KGs are more commonly utilized to expedite the drug re-purposing process (Zhu et al., 2020; MacLean, 2021; Himmelstein et al., 2017; Zhang et al., 2021b; Gao et al., 2022; Xu et al., 2019; Zhang & Che, 2021; Fang et al., 2023; Ghorbanali et al., 2023). Many applications on drug re-purposing that utilize KGs are primarily focused on link prediction tasks (MacLean, 2021). To re-purpose promising drug candidates for new indications, many methods employ predictive models that focus on predicting drug-treats-disease relationships within pharmacological knowledge graphs KGs. Himmelstein et al. (2017) use a degree-normalized pathway model on the hetionet KG which includes genes, diseases, tissues, pathophysiologies, and multimodal edges, to identify potentially repurposable drugs for epilepsy. Xu et al. (2019) develop a multi-path random walk model on a network that incorporates gene-phenotype associations, protein-protein interactions, and phenotypic similarities for training and prediction purposes. Zhang et al. (2021b) introduce an integrative and literature-based discovery model for identifying potential drug candidates from COVID-19-

focused research literature, including PubMed and other relevant sources. Gao et al. (2022) construct a knowledge graph (KG) by integrating multiple genotypic and phenotypic databases. They then learn low-dimensional representations of the KG and utilize these representations to infer new drug-disease interactions, providing insights into potential drug repurposing opportunities. Zhang & Che (2021) introduce a model for drug re-purposing in Parkinson's disease that leverages a local medical knowledge base incorporating accurate knowledge along with medical literature containing novel information. Ghorbanali et al. (2023) present the DrugRep-KG method, which utilizes a KG embedding approach for representing drugs and diseases in the process of drug repurposing.

### 4.2.2. CLINICAL TRIAL

The major goal of clinical trials is to assess the safety and effectiveness of drug molecules on human bodies. A novel drug molecule needs to pass three phases of clinical trials before it is approved by Food and Drug Administration (FDA) and enters the drug market. The whole process is prohibitively time-consuming and expensive, costing 7-11 years and two billion dollars on average (Martin et al., 2017).

**Clinical Trial Optimization** targets on identifying eligible patients for clinical trials based on their medical history and health conditions (Rivera et al., 2020; He et al., 2020). Recently, with massive electronic health records (EHR) data and trial eligibility criteria (EC), data-driven methods have been studied to automatically assign appropriate patients for clinical trials (Yuan et al., 2019; Tseo et al., 2020; Liu et al., 2021b). However, it is often hard to fully capture and represent the complex knowledge present in unstructured ECs and EHR data, as ECs may only provide general disease concepts. In contrast, patient EHR data contain more specific medical codes to represent patient conditions. To better capture the interactions among different medical concepts from EHR records and ECs, Gao et al. (2020a) enhance patient records with hierarchical taxonomies to align medical concepts of varying granularity between EHR codes and ECs. Besides, Fu et al. (2022) leverage additional knowledge-embedding modules along with drug pharmacokinetic and historical trial data to improve the patient trial optimization process, and Wang et al. (2023) leverage the knowledge graphs to learn static trial embedding and further designed meta-learning module to generalize well over the imbalanced clinical trial distribution.

### 4.3. Clinical Decision Support

Nowadays, abundant Electronic Health Record (EHR) data enables better computational models for accurate diagnoses and treatments. EHR contains essential patient information such as disease diagnoses, prescribed medications, and test results. Due to this valuable information, EHRs are extensively utilized to identify patterns in patient health and assist healthcare providers in making informed clinical decisions.

To facilitate automatic clinical predictions, various deep-learning-based approaches have been adopted including recurrent neural networks (RNN) (Choi et al., 2016; Ma et al., 2017; Yin et al., 2019; Fu et al., 2019; Raket et al., 2020; Gao et al., 2020b; Chen et al., 2019; Guo et al., 2021), graph neural networks (GNN) (Choi et al., 2020; Wang et al., 2021b; Zhu & Razavian, 2021; Xu et al., 2022; Nelson et al., 2022; Mao et al., 2022; Cai et al., 2022; Nikolentzos et al., 2023; Xu et al., 2023a), and transformers (Ma et al., 2020; Prakash et al., 2021; Antikainen et al., 2023; Labach et al., 2023; Chen et al., 2023b; Shickel et al., 2023). However, the sparsity of EHR data typically allows for only a small fraction of medical codes to be learned effectively, thereby restricting the ability of deep learning approaches. To overcome this drawback, KGs have been applied to incorporate prior medical knowledge for these deep learning models, which augment the representation of medical codes to better support the downstream tasks.

### 4.3.1. STEPS TO ADVANCE PREDICTION MODELS

**ICD Coding** aims to extract diagnosis and procedure codes from clinical notes which often consisted of raw text (Mullenbach et al., 2018; Zhang et al., 2020; Vu et al., 2021; Dong et al., 2022). It is often challenging, as the size of the candidate target codes can be large and their distribution is often long-tailed (Kim & Ganapathi, 2021). To overcome this, Xie et al. (2019) and Cao et al. (2020) propose to leverage knowledge graphs as *distant supervision* (Min et al., 2013; Zhang et al., 2022) and inject the label information via structured *knowledge graph propagation* by leveraging graph convolution networks to learn the correlations among medical codes. Besides, Lu et al. (2020) propose to leverage knowledge graphs as well as the co-occurrence graph among clinical nodes simultaneously with a knowledge aggregation module to boost the performance of ICD coding further. Overall, injecting additional knowledge with graph neural networks offers a way to mitigate the imbalanced label distribution and thus better.

**Entity and Relation Extraction from Health Records.** Health records contain rich unstructured or semi-structured data, making it difficult for clinicians to analyze relevant information. Entity and relation extraction helps convert this unstructured text into structured data that can be more easily processed, understood, and utilized. Specifically, Varma et al. (2021) transfer structural knowledge from the knowledge base to the medical domain, which improved the disambiguation accuracy of rare entities. Fries et al. (2021) leverage clinical ontologies to provide *weak supervision* sources to create additional training data for clinical entity disambiguation. Yuan et al. (2023) inject additional

knowledge from the knowledge graphs for entity linking, and proposed Post-pruning and thresholding techniques to reduce the effect of unlinkable entity mentions. Besides, Fei et al. (2021); Roy & Pan (2021) propose additional post-training steps to align the language models with biomedical knowledge. Hong et al. (2021) construct embeddings for a wide range of codified concepts from EHRs to identify relevant features related to a disease of interest, and Lin et al. (2022) design a co-training scheme to jointly learn from text and knowledge graphs for extracting disease-disease relations. Fusing knowledge graphs with deep language models can flexibly accommodate missing data types and brings additional performance gains, especially for those rare entities and relations.

**Clinical Report Summarization.** Numerous studies have focused on transforming raw patient visit data into concise yet informative medical reports to enhance the automatic diagnosis process. Although standard models for medical report generation have achieved promising performances, there is usually no guarantee of the clinical informativeness of the generated text. To improve the faithfulness of the summarized text, Biswal et al. (2020) exploit the anchor words of relevant disease phenotypes from the external knowledge base to ensure the clinical accuracy of the generated report. Liu et al. (2021a) use an additional memory-augmented module to distill the fine-grained knowledge preserved in the knowledge graph to acquire accurate report generation. Besides, another specific issue of the clinical report is the *missing data*: some attributes are inevitably missing (Cismondi et al., 2013) when scored by domain experts. To combat the missing data issue, Xi et al. (2021) design a knowledge-aware encoder-decoder structure that injects structural information from knowledge graphs during the encoding stage and infers patients' links to clinical outcomes during the decoder stage. In summary, incorporating external knowledge graphs into clinical report summarization enhances the content's factual accuracy and alleviates the impact of missing data by grounding the generated text in verified knowledge.

### 4.3.2. EVIDENCE GENERATION FOR RISK PREDICTION

**Disease Prediction** aims to predict the potential diseases of a given patient with his past clinical records. To assist the diagnosis with additional knowledge, GRAM (Choi et al., 2017) and KAME (Ma et al., 2018) utilize a medical ontology (Dubberke et al., 2006) where the leaf nodes are the medical codes found in EHR data, and their ancestors are more general categories. By incorporating information from medical ontologies into deep learning models via neural attention, these approaches learn better embeddings for different medical concepts to alleviate the data scarcity bottleneck. Yin et al. (2019); Zhang et al. (2019) further consider the domain-specific knowledge graph KnowLife (Ernst et al.,

2015) to enrich the embeddings of medical entities with their neighbors on the knowledge graph. These approaches mainly directly update the embeddings of different concepts to improve feature learning, but ignore the high-level order information from the knowledge graph. To tackle this drawback, Ye et al. (2021a) explicitly exploit *paths* in KG from the observed symptoms to the target disease to model the personalized information for diverse patients with a relational-guided attention mechanism. Xu et al. (2021) design a self-supervised learning approach to pretrain a graph attention network for learning the embedding of medical concepts and completing the knowledge graph simultaneously. These approaches better harness the structure information, and often lead to better performance than the pure embedding-based knowledge integration techniques.

**Treatment Recommendation** aims to recommend personalized medications to patients based on their individual health conditions, which can help physicians to select the most effective medications for their patients, and improve treatment outcomes (Zhang et al., 2017; Bhoi et al., 2021; Shang et al., 2019b). To effectively exploit external knowledge, Shang et al. (2019a) use drug ontologies to design additional pretraining loss and directly improve the representation of drugs, and several studies (Wu et al., 2022; Tan et al., 2022; Yang et al., 2021) attempt to extract the additional drug interaction graphs to model the negative side effects of specific drug pairs and reduce the possibility of recommending negative drug-drug interaction combinations.

## 5. Promise and Outlook

Several promising research directions in computer science are poised to generate large and accurate healthcare knowledge graphs (HKGs) in the near future. One such direction involves the development of advanced entity and relation extraction techniques that can effectively capture and represent complex biomedical knowledge. Additionally, large language models (*a.k.a.* foundation models) like GPT-3 have demonstrated promise in capturing the semantics and context of biomedical language (Agrawal et al., 2022; Singhal et al., 2022; Nath et al., 2022), allowing researchers to better understand and interpret complex biomedical data (Moor et al., 2023). Another direction includes the use of graph-based learning paradigms capable of effectively integrating heterogeneous data sources and learning from the complex relationships within HKGs. These approaches can facilitate the creation of comprehensive and fine-grained HKGs that capture a diverse range of biomedical knowledge, ultimately promoting the interpretability of biomedical research and clinical decision-making.

The potential impact of comprehensive and fine-grained HKGs on biomedical research and clinical practice is significant. By integrating vast amounts of biomedical knowledge

from multiple domains, HKGs can facilitate the discovery of new disease mechanisms and identification of novel drug targets. They also hold the potential to enable personalized medicine by identifying patient subgroups with shared disease mechanisms and guiding the selection of targeted interventions based on individual patient characteristics. Additionally, HKGs can enhance clinical decision-making by providing access to up-to-date and pertinent biomedical knowledge, thereby improving efficiency and accuracy. Overall, the creation and utilization of HKGs present a promising avenue for promoting transparency and interpretability for clinical practice.

## 6. Conclusion

In conclusion, healthcare knowledge graphs (HKGs) offer a promising approach to capturing and organizing medical knowledge in a structured and interpretable way, providing a comprehensive and fine-grained view of medical concepts and relationships. Despite challenges like data heterogeneity and limited coverage, recent technical advancements have enabled the creation of comprehensive and precise HKGs. This survey provides a comprehensive overview of the current state of HKGs, covering their construction, utilization models, and applications in healthcare. We also discuss potential future developments, emphasizing the importance of HKGs in facilitating efficient and effective healthcare delivery. With the emergence of large language models, the potential for creating even more comprehensive and precise HKGs is unprecedented. In conclusion, healthcare knowledge graphs (HKGs) hold great potential in improving the interpretability of healthcare, enabling transparent and informed decision-making, and promoting evidence-based practices for better patient outcomes.

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
