# OpenReview forum: "A Survey on Knowledge Graphs for Healthcare: Resources, Application Progress, and Promise"
_ICML.cc/2023/Workshop/IMLH — IMLH 2023 Poster_

### Official Review · Reviewer_3Ua2 · 2023-06-12
**A Survey on Knowledge Graphs for Healthcare: Resources, Application Progress, and Promise**

**Rating:** 6
**Confidence:** 4

**Review:**

In this paper, the authors present a survey on healthcare knowledge graphs (HKGs). The key HKG construction techniques, resources, application progress and promise are summarised in this survey. The application progress is well-written. However, the following comments need to be considered.

1.	This paper claims “this survey paper represents the first comprehensive overview of healthcare knowledge graphs (HKGs)”. But, an HKG review [1] has been published recently. The authors should introduce this related work and highlight their differences.
[1] Abu-Salih, Bilal, et al. "Healthcare knowledge graph construction: A systematic review of the state-of-the-art, open issues, and opportunities." Journal of Big Data 10.1 (2023): 81.
2.	In the resources section, only Table 1 is provided. The limited resources are listed. Moreover, there are some ontologies that are not typical knowledge graphs in the listed resources. More knowledge graph resources should be provided, e.g., PharmKG [2], KGHC [3], and so on.
[2] Zheng, Shuangjia, et al. "PharmKG: a dedicated knowledge graph benchmark for biomedical data mining." Briefings in bioinformatics 22.4 (2021): bbaa344.
[3] Li, Nan, et al. "KGHC: a knowledge graph for hepatocellular carcinoma." BMC Medical Informatics and Decision Making 20.3 (2020): 1-11.

---

### Official Review · Reviewer_hPQU · 2023-06-16
**A comprehensive, well-structured, and useful survey on knowledge graph for healthcare**

**Rating:** 7
**Confidence:** 4

**Review:**

This paper provides a comprehensive overview of the current state of HKGs, their construction techniques, utilization models, and applications in healthcare.

- This survey is well-structural, well-written and detailed;

- KG is an important topic and useful tool in Healthcare, so this survey might be valuable resource for researchers in this area;

- It would be beneficial to include more insights on how knowledge graphs can contribute to enhancing model explainability in healthcare applications.

- It would be valuable for the authors to include a more thorough discussion on the limitations and challenges faced in current research.

---

### Official Review · Reviewer_ENEe · 2023-06-18
**Good Paper**

**Rating:** 8
**Confidence:** 5

**Review:**

# Review

Paper type: 8-page long paper.

## Summary

This paper presents a comprehensive survey for the knowledge graph in healthcare (HKG). It summarizes the pipeline and key techniques for HKG construction, utilization as well as the resources available to develop and apply HKG. The paper also conducts a comprehensive study on the applications of HKGs, which ranges from basic science research, pharmaceutical research, to clinical decision support. Finally, the paper discusses the potential opportunities of HKGs with the emerging technology like large language models that can revolutionize healthcare and enhance the interpretability and reliability of clinical prediction.

## Strength

This paper is the first to provide a comprehensive survey of the use of knowledge graphs in healthcare. Compared with the existing survey of the knowledge graph, such as

1. S. Ji, S. Pan, E. Cambria, P. Marttinen and P. S. Yu, "A Survey on Knowledge Graphs: Representation, Acquisition, and Applications," in IEEE Transactions on Neural Networks and Learning Systems, vol. 33, no. 2, pp. 494-514, Feb. 2022, doi: 10.1109/TNNLS.2021.3070843.
2. Abu-Salih, Bilal, et al. "Healthcare Knowledge Graph Construction: State-of-the-art, open issues, and opportunities." arXiv preprint arXiv:2207.03771 (2022).
3. Hogan, Aidan, et al. "Knowledge graphs." ACM Computing Surveys (CSUR) 54.4 (2021): 1-37.
4. Wang, Quan, et al. "Knowledge graph embedding: A survey of approaches and applications." IEEE Transactions on Knowledge and Data Engineering 29.12 (2017): 2724-2743.

This paper is complementary to these surveys in that: (1) It focuses on the context of healthcare, using the original biological terms to describe usage of HKG; (2) it provides direct and recent applications of the knowledge graph in healthcare and (3) it highlights the opportunities of the knowledge graph with the emerging technologies.

The entire paper is well structured and easy (and pleasant) to read. The headings and Figure 1 are helpful to understand the structure of the paper. The literature cited in this paper is relevant and up-to-date.

## Weakness

I had attempted to find a weakness of the paper, but it is hard to find one. The paper is well-organized and has no major structural issues. One suggestion is to emphasize the role of HKG in clinical trials in section 4.2.2, as it is not obvious until section 4.3. Another grammatical suggestion is to remove the unnecessary comma at line 042.

## Rating and Justification

Acceptance. This survey paper summarizes well the usage of the knowledge graph in healthcare. It also discusses promising research directions with emerging technology like large language models and new graph based learning methods. HKG has the potential to “improve the interpretability of healthcare, enable transparent and informed decision-making in clinics, and promote evidence-based practices for better patient outcomes”. I believe that this paper would be a valuable resource for future researchers in the field.

---

### Official Review · Reviewer_1k1Q · 2023-06-18
**This survey serves as an overview of HKGs, summarizing their construction techniques, utilization approaches, and transformative impact across various healthcare domains.**

**Rating:** 6
**Confidence:** 2

**Review:**

I am not sure about how to evaluate a survey in a workshop, especially when I am not familiar with the topic of KG.

---

### Meta-Review · Area_Chair_Lp9v · 2023-06-20

**Recommendation:** Accept (Poster)
**Confidence:** 3

**Metareview:**

This work provides a comprehensive survey of knowledge graphs in healthcare. The topic is suitable, and there is no identified flaw in constructing and presenting the survey. The authors should provide a comparison of this survey with existing surveys, as mentioned by the reviewer, and incorporate other reviewers' comments in their revision of the survey.

---

### Decision · Program_Chairs · 2023-06-20

Accept (Poster)